# Association between serum ferritin level and decreased diffusion capacity 3 months after the onset of COVID-19 pneumonia

Kyota Shinfuku[1]*, Naoki Takasaka[1], Taiki Fukuda[2], Kentaro Chida[1], Yudai Suzuki[1],
Shun Shibata[1], Ayako Kojima[1], Tsukasa Hasegawa[1], Masami Yamada[1],
Yumie Yamanaka[1], Yusuke Hosaka[1], Aya Seki[1], Yoshitaka Seki[1], Hiroshi Takeda[1],
Takeo Ishikawa[1], Kazuyoshi Kuwano[3]

1 Division of Respiratory Diseases, Department of Internal Medicine, The Jikei University Daisan Hospital,
Tokyo, Japan, 2 Department of Radiology, The Jikei University Daisan Hospital, Tokyo, Japan, 3 Division of
Respiratory Diseases, Department of Internal Medicine, The Jikei University School of Medicine, Tokyo,
Japan

* shinfuku.112@gmail.com

pone.0281249

Sant'Anna, SWITZERLAND

**Data Availability Statement:** All relevant data are
within the paper.

## Abstract

### Background

Coronavirus disease 2019 (COVID-19) pneumonia can have prolonged sequelae and lead
to respiratory dysfunction, mainly because of impaired diffusion capacity for carbon monox-
ide (DLCO). The clinical factors associated with DLCO impairment, including blood bio-
chemistry test parameters, remain unclear.

### Methods

Patients with COVID-19 pneumonia who underwent inpatient treatment between April 2020
and August 2021 were included in this study. A pulmonary function test was performed 3
months after onset, and the sequelae symptoms were investigated. Clinical factors, includ-
ing blood test parameters and abnormal chest shadows on computed tomography, of
COVID-19 pneumonia associated with DLCO impairment were investigated.

### Results

In total, 54 recovered patients participated in this study. Twenty-six patients (48%) and 12
patients (22%) had sequelae symptoms 2 and 3 months after, respectively. The main
sequelae symptoms at 3 months were dyspnea and general malaise. Pulmonary function
tests showed that 13 patients (24%) had both DLCO <80% predicted value (pred) and
DLCO/alveolar volume (VA) <80% pred, and appeared to have DLCO impairment not attrib-
utable to an abnormal lung volume. Clinical factors associated with impaired DLCO were
investigated in multivariable regression analysis. Ferritin level of >686.5 ng/mL (odds ratio:
11.08, 95% confidence interval [CI]: 1.84–66.59; $p$ = 0.009) was most strongly associated
with DLCO impairment.

**Funding:** The author(s) received no specific funding for this work.

**Competing interests:** The authors have declared that no competing interests exist.

## Conclusions

Decreased DLCO was the most common respiratory function impairment, and ferritin level was a significantly associated clinical factor. Serum ferritin level could be used as a predictor of DLCO impairment in cases of COVID-19 pneumonia.

## Introduction

The coronavirus disease (COVID-19) pandemic has resulted in more than 614 million confirmed cases with more than 6.5 million deaths as of October 2, 2022 [1]. Owing to the development of vaccines, infections are declining in areas where vaccination coverage is high. However, the uneven distribution of vaccines by region and the emergence of mutant strains of severe acute respiratory syndrome coronavirus 2 (SARS-CoV-2) has caused the pandemic to persist. SARS-CoV-2 infects the upper respiratory tract, bronchiolar epithelial cells, alveolar epithelial cells, alveolar macrophages, and vascular endothelium via angiotensin-converting enzyme 2 [2]. Excessive immune response and the infecting virus dose are thought to be major factors contributing to the severity of the disease [3]. Dexamethasone, tocilizumab, and baricitinib can be used to suppress the excessive immune response.

In addition, the sequelae after COVID-19 are often prolonged and persist even after viral excretion is stopped. At 60 days after discharge, 87.4% of patients were reported to have symptoms of sequelae, primarily general malaise and dyspnea [4]. In patients who have recovered from COVID-19, reduced diffusion capacity for carbon monoxide (DLCO) has been reported as one of the most frequent respiratory dysfunctions [5, 6]. This abnormality has also been observed in survivors of severe acute respiratory syndrome (SARS) and Middle East respiratory syndrome (MERS) [7, 8]. Reduced DLCO has been reported in COVID-19 survivors at the time of discharge and various times thereafter. Mo et al. [9] reported that 47% of COVID-19 survivors showed DLCO impairment at discharge. Previous studies have reported that DLCO impairment remained in 16–82% of cases 3 months after discharge [5, 10, 11]. The severity of COVID-19 and the spread of abnormal chest shadows on computed tomography (CT) scans in COVID-19 pneumonia cases have been reported to be associated with DLCO reduction [12–14]. D-dimer has been reported as a blood biochemical marker [5]. The clinical factors associated with DLCO impairment after COVID-19 pneumonia have not yet been fully clarified. The purpose of this study was to investigate sequelae symptoms and respiratory dysfunction after hospitalization for COVID-19 pneumonia and to identify clinical factors associated with DLCO impairment.

## Methods

### Study design

This retrospective study was conducted in patients who received inpatient treatment between April 2020 and August 2021 at the Jikei University Daisan Hospital. All COVID-19 diagnoses were based on SARS-CoV2 by real-time reverse transcriptase-polymerase chain reaction, transcription reverse-transcription concerted reaction, or antigen testing. Enrolled patients were those who were hospitalized with COVID-19 pneumonia during the period and who requested outpatient follow-up after the sequelae symptoms and respiratory dysfunction were explained to them. From the standpoint of infection control, an outpatient examination and CT scan were performed 2 months later and it was confirmed that the patient was improving. Then, pulmonary function tests were performed 3 months later. Severity was defined according to

the COVID-19 treatment guidelines of the National Institute of Health [15]. In this study, patients were classified into three groups according to the aforementioned guidelines. Groups 1, 2, and 3 included patients who were hospitalized but did not require oxygen supplementation; those who were hospitalized and required conventional oxygen; and those who were hospitalized and required high-flow nasal cannula, noninvasive ventilation, invasive mechanical ventilation, or extracorporeal membrane oxygenation, respectively. Data were collected on patients' clinical characteristics, blood test results, pulmonary function test, and chest CT images.

## Blood test results

Blood tests were performed on admission and periodically thereafter throughout hospitalization. Then, we evaluated the blood sample data during the hospitalization period that may be related to the severity of COVID-19. The peak values of white blood cells, aspartate aminotransferase (AST), alanine aminotransferase (ALT), total-bilirubin, creatine, lactate dehydrogenase (LDH), C-reactive protein, erythrocyte sedimentation rate, ferritin, KL-6, and D-dimer, as well as the minimum values for platelets and hemoglobin, were recorded [16].

## Pulmonary function test (PFT)

PFT was performed 3 months after onset by a professional inspection engineer using a FUDAC-7 PFT system (Fukuda Denshi Co. Ltd., Tokyo, Japan). The recorded parameters included the following: total lung capacity (TLC), forced vital capacity (FVC), residual volume, forced expiratory volume in the first second ($FEV_1$), $FEV_1$/FVC ratio, DLCO, and DLCO/alveolar volume (VA). In this study, cases with both DLCO <80% predicted value (pred) and DLCO/VA <80% pred were defined as true DLCO impairment, unaffected by lung volume.

## Chest CT

Chest high-resolution CT was performed on all patients using a helical scanner (SOMATOM Definition AS +, Siemens, Erlangen, Germany) with a 1.0-mm slice thickness. Participants underwent chest CT on admission and in the outpatient department 2 months after onset. One radiologist and two pulmonologists, all with more than 10 years of experience, reviewed CT images without knowledge of the patient's clinical information. For each evaluated CT image, differences in evaluations were settled by consensus. The major CT findings were described using standard international nomenclature defined by the Fleischer Society glossary using terms including ground-glass opacities, consolidation, and reticular pattern [17]. A semiquantitative scoring was based on the chest CT total severity score (TSS) according to the percentage involvement of these abnormalities in each lobe [18, 19]. Each of the five lung lobes was scored on a 5-point scale as follows: 0, indicating no involvement; 1, <25% involvement; 2, 26–49% involvement; 3, 50–75% involvement; and 4, >75% involvement. TSS was the sum of the individual lobar scores and ranged from 0 to 20 points. In addition, the presence or absence of fibrosis on CT image was evaluated on chest CT examination performed at 2 months after onset, based on the presence of traction, parenchymal bands, or honeycombing [20]. Fibrosis was defined as present according to the CT findings when there were two or more of these findings.

## Statistical analysis

Data were presented as number (percentage), mean (standard deviation, SD), or median (interquartile range, IQR) depending on whether they were normally or non-normally

distributed based on the Shapiro-Wilk test. Fisher's exact test or the chi-square test was used to compare categorical values between groups, and t-tests or the Mann-Whitney U test and ANOVA or the Kruskal-Wallis test were used to compare continuous variables, depending on whether the data were parametric or non-parametric, respectively. Receiver operating characteristic (ROC) curve analysis and area under the curve (AUC) were used to assess the risk factors associated with DLCO impairment. The cutoff value for each risk factor was determined using Youden's Index. Multivariable logistic regression models were used to examine the association between clinical factors with cutoff values and DLCO impairment. The statistical analyses were performed using GraphPad Prism version 8.4.3 for Macintosh (GraphPad Software La Jolla, CA, USA) and SPSS version 24 (IBM Corp., Armonk, NY, USA), and the statistical significance was set at $p < 0.05$.

### Ethics statement

The study was conducted in accordance with the Declaration of Helsinki. The protocol was approved by the Ethics Committee of Jikei University School of Medicine [No. 30-003(9024)]. In accordance with the ethical guidelines of the Jikei University School of Medicine, informed consent was not necessary for this retrospective study, and opt-out consent was provided on the university website.

## Results

### Patient characteristics and symptoms of sequelae

From April 2020 to August 2021, 362 patients were admitted to our hospital. Of these, 54 patients who underwent outpatient examinations were included in this study (Table 1). The mean (SD) age was 58.8 (14.8) years, and 35 patients (65%) were male. In comparison between the three groups, age was found to be higher in the oxygen administration groups ($p < 0.001$). All patients were Asian by race. The major complications were hypertension (19 patients, 35%), diabetes mellitus (12 patients, 22%), and respiratory diseases (6 patients, 11%). Respiratory diseases included well-controlled bronchial asthma, pneumothorax, and sleep apnea syndrome, and were not considered to significantly affect PFTs.

Two months after onset, 26 patients (48%) were left with sequelae, and 3 months after onset, 12 patients (22%) had residual symptoms. Details of the symptoms are shown in Table 1. The main sequelae symptoms at 3 months were dyspnea and general malaise. The frequency of sequelae at both 2 and 3 months was higher in group 3 than in the other groups ($p = 0.014$ and $p = 0.027$, respectively).

### Pulmonary function 3 months after COVID-19 onset

Pulmonary function tests 3 months after onset showed a decrease in TLC <80% pred in 4 patients (7%), restrictive impairment in FVC <80% in 4 patients (7%), and obstructive impairment in FEV 1.0/FVC <70% in 5 patients (9%). Twenty-eight patients (52%) had DLCO <80% pred and 20 patients (37%) had DLCO/VA <80% pred. There were 13 patients (24%) who fulfilled both DLCO <80% pred and DLCO/VA <80% pred, and had DLCO impairment unaffected by lung volume abnormality (Table 1).

Comparing pulmonary function according to severity, TLC % pred and FVC % pred were significantly lower in patients with more severe disease. With regard to diffusion impairment, DLCO % pred was lower in group 3, but the difference was not statistically significant. DLCO/VA decreased significantly according to disease severity ($p = 0.004$), and the proportion of patients who met both DLCO <80% pred and DLCO/VA <80% pred and had DLCO

**Table 1. Characteristics of patients and comparison by severity.**

| Parameters | Total (n=54) | Group1: Hospitalized but dose not requires oxygen supplementation (n=25) | Group2: Hospitalized and require conventional oxygen (n=22) | Group 3: Hospitalized and requires HFNC, NIV or IMV, ECMO (n=7) | p-value |
|---|---|---|---|---|---|
| **Age (years) [a]** | **58.8 ± 14.8** | **50.5 ± 13.7** | **67.5 ± 12.5** | **60.8 ± 8.7** | **<0.001** |
| Sex, (%Male) [b] | 35 (65) | 13 (52) | 15 (68) | 7 (100) | 0.057 |
| Race [b] | | | | | |
| Asian | 54 (100) | 25 (100) | 22 (100) | 7 (100) | NA |
| BMI (kg/m$^2$) [a] | 25.2 ± 4.1 | 26.1± 4.6 | 24.7 ± 3.9 | 23.4 ± 2.1 | 0.259 |
| Smoking history [b] | 25 (46) | 11 (44) | 11 (50) | 3 (43) | 0.904 |
| Comorbidities [b] | | | | | |
| **Hypertension** | **19 (35)** | **2 (8)** | **14 (64)** | **3 (43)** | **<0.001** |
| Diabetes melitus | 12 (22) | 2 (8) | 8 (36) | 2 (29) | 0.059 |
| Respiratory diseases † | 6 (11) | 4 (16) | 1 (5) | 1 (14) | 0.441 |
| Clinical course [c] | | | | | |
| **Hospital period (days)** | **13.5 (10-17)** | **11 (9-13.5)** | **14 (13-17)** | **24 (17-46)** | **<0.001** |
| **Frequency of blood test (days apart)** | **2.6 (2.2-3)** | **3 (2.5-3.1)** | **2.5 (2.3-2.8)** | **1.7 (1.6-2.2)** | **<0.001** |
| Time from onset to examination [a] | | | | | |
| Chest CT after discharge (days) | 51.9 ± 8.6 | 48.8 ± 7.7 | 51.7 ± 5.0 | 63.2 ± 11.9 | 0.002 |
| Lung function (days) | 95.3 ± 15.5 | 97 ± 18.1 | 94.4 ± 13.6 | 92.2 ± 12.3 | 0.851 |
| Sequelae symptoms [b] | | | | | |
| **after 2 months** | **26 (48)** | **14 (56)** | **6 (27)** | **6 (86)** | **0.014** |
| dyspnea | 13 (24) | 7 (28) | 3 (14) | 3 (42) | |
| cough•sputum | 5 (9) | 2(8) | 1 (5) | 2 (29) | |
| general malaise | 4 (7) | 1 (4) | 2 (9) | 1(14) | |
| hair loss | 2 (4) | 2 (8) | 0 | 0 | |
| memory loss | 1 (2) | 1(4) | 0 | 0 | |
| chest tightness | 1 (2) | 1(4) | 0 | 0 | |
| **after 3 months** | **12 (22)** | **6 (24)** | **2 (9)** | **4 (57)** | **0.027** |
| dyspnea | 5 (9) | 2 (8) | 0 | 3 (43) | |
| cough•sputum | 0 | 0 | 0 | 0 | |
| general malaise | 3 (6) | 1 (4) | 1(5) | 1 (14) | |
| hair loss | 2 (4) | 2 (8) | 0 | 0 | |
| memory loss | 1 (2) | 0 | 1(5) | 0 | |
| chest tightness | 1 (2) | 1(4) | 0 | 0 | |
| Lung function | | | | | |
| Lung volume | | | | | |
| **TLC % pred [a]** | **104.4 ± 15.74** | **109.6 ± 15.85** | **103.1 ± 11.33** | **90.07 ± 19.46** | **0.010** |
| **TLC < 80% pred [b]** | **4 (7)** | **1 (4)** | **0 (0)** | **3 (43)** | **<0.001** |
| RV % pred [a] | 107 ± 24.94 | 106.3 ± 28.5 | 108.9 ± 21.95 | 103.4 ± 23.05 | 0.870 |
| Spirometry | | | | | |
| **FVC % pred [a]** | **106.6 ± 16.75** | **111.8 ± 16.16** | **104.4 ± 13.92** | **94.44 ± 21.23** | **0.035** |
| FVC < 80% pred [b] | 4 (7) | 1 (4) | 1 (5) | 2 (29) | 0.072 |
| **FEV1.0% pred [a]** | **106.1 ± 17.83** | **101.6 ± 15.2** | **113.2 ± 17.93** | **100.1 ± 21.08** | **0.048** |
| FEV1.0% < 80% pred [b] | 4 (7) | 1 (4) | 1 (5) | 2 (29) | 0.072 |
| FEV1.0 / FVC [a] | 79.2 ± 6.59 | 78.02 ± 7.25 | 79.32 ± 6.18 | 83.04 ± 4.03 | 0.206 |
| FEV1.0/FVC <70% [b] | 5 (9) | 2 (8) | 3 (14) | 0 (0) | 0.531 |
| Diffusion capacity | | | | | |
| DLCO % pred [a] | 81.56 ± 22.54 | 80.79 ± 17.01 | 87.75 ± 25.1 | 64.86 ± 25.63 | 0.061 |

(*Continued*)

**Table 1.** (Continued)

| Parameters | Total (n=54) | Group1: Hospitalized but dose not requires oxygen supplementation (n=25) | Group2: Hospitalized and require conventional oxygen (n=22) | Group 3: Hospitalized and requires HFNC, NIV or IMV, ECMO (n=7) | p-value |
|---|---|---|---|---|---|
| DLCO < 80% pred [b] | 28 (52) | 14 (56) | 9 (41) | 5 (71) | 0.316 |
| DLCO/VA % pred [a] | 90.53 ± 16.43 | 97.96 ± 14.92 | 85.55 ± 13.26 | 79.66 ± 20.47 | 0.004 |
| DLCO/VA < 80% pred [b] | 20 (37) | 4 (16) | 12 (55) | 4 (57) | 0.012 |
| DLCO < 80% pred and DLCO/VA < 80% pred [b] | 13 (24) | 3 (12) | 6 (27) | 4 (57) | 0.042 |

Data are presented as

[a] mean ± SD

[b] n (%), or [c] median (range quartile), NA: not applicable.

†: All four group 1 cases were of bronchial asthma; In group 2, there was one case of pneumothorax, whereas, in group 3, there was one case of sleep apnea syndrome. HFNC: high-flow nasal cannula, NIV: noninvasive ventilation, IMV: invasive mechanical ventilation, ECMO: extracorporeal membrane oxygenation, BMI: body mass index, TLC: total lung capacity, RV: residual volume, FVC: forced vital capacity, FEV1.0: forced expiratory volume in the first second. DLCO: diffusion capacity for carbon monoxide, VA: alveolar volume.

impairment not attributable to an abnormal lung volume was significantly higher in patients with more severe disease ($p$ = 0.042) (Table 1).

## Clinical factors related to DLCO impairment

The risk factors associated with DLCO impairment are shown in Table 2. Regarding patient characteristics, the prevalence of smoking history was higher in the DLCO-impaired group ($p$ = 0.023). Regarding clinical course and treatment, the rate of mechanical ventilator use was higher in the DLCO-impaired group ($p$ = 0.049). Regarding the laboratory test results, serum AST ($p$ = 0.034), LDH ($p$ = 0.024), and ferritin ($p$ = 0.001) level were significantly higher in the DLCO-impaired group. With regard to the chest CT findings, both TSS on admission ($p$ = 0.032) and at 2 months ($p$ = 0.005) were higher in the DLCO-impaired group. The prevalence of fibrosis was significantly higher in the DLCO-impaired group ($p$ = 0.005).

## ROC analysis of risk factors associated with DLCO impairment

We then used the ROC curve to examine the AUC and cutoff values by Youden Index for clinical factors that were found to be different in the univariate analysis. Table 3 and Fig 1 show the AUCs and cutoff values for the clinical factors. The AUC for ferritin level was highest: it was 0.783 (95% confidence interval [CI]: 0.616–0.949; $p$ = 0.002). The cutoff value was 686.5 ng/mL with a sensitivity of 84.6% and specificity of 75%. The AUCs for AST and LDH were lower, with cutoff values of 49.5 U/L and 273.5 U/L, respectively. The ROC analysis showed that the cut-offs for TSS at admission and after 2 months were 6.5 and 5.5, respectively, and the AUC was higher for TSS after 2 months, at 0.745 (95% CI: 0.566–0.924; $p$ = 0.008), than that at admission. The AUC for the presence of fibrosis at 2 months was 0.734 (95% CI: 0.569–0.899; $p$ = 0.012).

## Clinical factors associated with DLCO impairment by logistic regression

To examine clinical factors during inpatient care associated with DLCO impairment, multivariable logistic regression analysis adjusted for age, sex, and smoking history was performed for AST, LDH, ferritin, and TSS (on admission) using the cutoff values calculated using ROC curves (Table 4). The results showed that ferritin >686.5 ng/mL (odds ratio: 11.08, 95% CI: 1.84–66.59; $p$ = 0.009) was most strongly associated with DLCO impairment.

**Table 2. Characteristics of patients in the DLCO normal group and DLCO-impaired group 3 months after onset.**

| Parameters | DLCO normal group (n=41) | DLCO impaired group (n=13) | *p*-value |
|---|---|---|---|
| **Age (years)** [a] | 59.2 ±15.7 | 57.2 ±12.1 | 0.524 |
| Sex, (%Male) [b] | 25 (59) | 11 (85) | 0.107 |
| BMI (kg/m$^2$) [a] | 25.1± 4.2 | 25.2 ± 3.9 | 0.982 |
| **Smoking history** [b] | **15 (37)** | **10 (77)** | **0.023** |
| Time from onset to examination [a] | | | |
| Lung function (days) | **95.5 ± 17.1** | **94.7 ± 9.8** | **0.737** |
| **Severity** [b] | | | |
| Group 1 | **22 (54)** | **3 (23)** | **0.042** |
| Group 2 | **16 (39)** | **6 (46)** | |
| Group 3 | **3 (7)** | **4 (31)** | |
| Comorbidity [b] | | | |
| Hypertension | 15 (37) | 4 (31) | >0.999 |
| Diabetes melitus | 9 (22) | 3 (23) | >0.999 |
| Respiratory diseases | 5 (12) | 1 (8) | >0.999 |
| Sequelae symptoms [b] | | | |
| after 2 months | 20 (49) | 6 (46) | >0.999 |
| after 3 months | 8 (20) | 4 (31) | 0.452 |
| Clinical course and treatment | | | |
| Hospital period (days) [c] | 13 (10-16) | 15 (11-19.5) | 0.143 |
| Dexamethasone [b] | 11 (27) | 4 (31) | >0.999 |
| Remdesivir [b] | 7 (17) | 4 (31) | 0.429 |
| Heparine [b] | 4 (10) | 2 (15) | 0.622 |
| **Mechanical ventilator use† [b]** | **3 (7)** | **4 (31)** | **0.049** |
| Laboratory data | | | |
| WBC (/μL) [c] | 7200 (5300-9000) | 8300 (7200-10400) | 0.079 |
| Platelets (×10$^4$/μL) [c] | 15.2 (12.95-19.05) | 16.8 (13.05-20.7) | 0.585 |
| Hemoglobin (/μL) [c] | 13.50 (11.8-14.6) | 13.1 (11.75-13.85) | 0.385 |
| **AST (U/L)** [c] | **39 (29.5-64.5)** | **61 (42-124)** | **0.034** |
| ALT (U/L) [c] | 45 (29-72.5) | 98 (40-154) | 0.054 |
| T-Bil (mg/dL) [c] | 0.6 (0.5-0.85) | 0.75 (0.6-0.97) | 0.145 |
| Cr (mg/dL) [a] | 0.93 ± 0.25 | 1.04 ± 0.26 | 0.225 |
| **LDH (U/L)** [c] | **266.5 (225.5-357)** | **379 (288.5-529)** | **0.024** |
| CRP (mg/dL) [c] | 4.61 (1.87-10.42) | 8.16 (4.89-12.59) | 0.151 |
| ESR (mm/hr) [a] | 33.72 ± 20.44 | 38.64 ± 22.07 | 0.496 |
| **Ferritin (ng/mL)** [c] | **407 (178.5-692.5)** | **1024 (754.5-2258)** | **0.001** |
| KL-6 (U/mL) [c] | 243 (192-288) | 244.5 (165.5-820) | 0.674 |
| D-dimer (μg/mL) [c] | 1.1 (0.8-1.7) | 1.2 (0.8-3.5) | 0.337 |
| Evaluation of chest CT | | | |
| **TSS (admission)** [c] | **5 (3-5.5)** | **6 (4.5-10)** | **0.032** |
| **TSS (2months after onset)** [c] | **3 (0-5)** | **6 (3.5-8)** | **0.005** |
| **Presence of fibrosis (2months after onset)** [c] | **9 (21.9)** | **9 (69.2)** | **0.005** |

Data are presented as [a] mean ± SD, [b] n (%), or [c] median (range quartile).

†: Mechanical ventilator includes high-flow nasal cannula, noninvasive ventilation, invasive mechanical ventilation, and extracorporeal membrane oxygenation.

BMI: body mass index, WBC: white blood cell, AST: aspartate aminotransferase, ALT: alanine aminotransferase, T-Bil: total-bilirubin, Cr: creatinine. LDH: lactate dehydrogenase, CRP: C-reactive protein, ESR: erythrocyte sedimentation rate, CT: computed tomography, TSS: total severity score.

**Table 3. ROC curve and cut-off value for each risk factor.**

| Risk factors | AUC (CI) | *p*-value | cut off value | sensitivity | specificity |
|---|---|---|---|---|---|
| AST (U/L) | 0.689 (0.519-0.859) | 0.042 | 49.5 | 0.769 | 0.650 |
| LDH (U/L) | 0.702 (0.537-0.867) | 0.030 | 273.5 | 0.923 | 0.500 |
| Ferritin (ng/mL) | 0.783 (0.616-0.949) | 0.002 | 686.5 | 0.846 | 0.750 |
| TSS (admission) | 0.687 (0.499-0.874) | 0.045 | 6.5 | 0.462 | 0.875 |
| TSS (2months after onset) | 0.745 (0.566-0.924) | 0.008 | 5.5 | 0.538 | 0.925 |
| Presence of fibrosis (2months after onset) | 0.734 (0.569-0.899) | 0.012 | ••• | 0.692 | 0.780 |

ROC: receiver operating characteristic, AUC: area under the curve, CI: confidence interval, AST: aspartate aminotransferase, LDH: lactate dehydrogenase, TSS: total severity score.

## Discussion

In this study, among 54 inpatients with COVID-19, 26 (48%) and 12 (22%) had sequelae symptoms after 2 and 3 months, respectively. The most frequent symptoms were dyspnea and

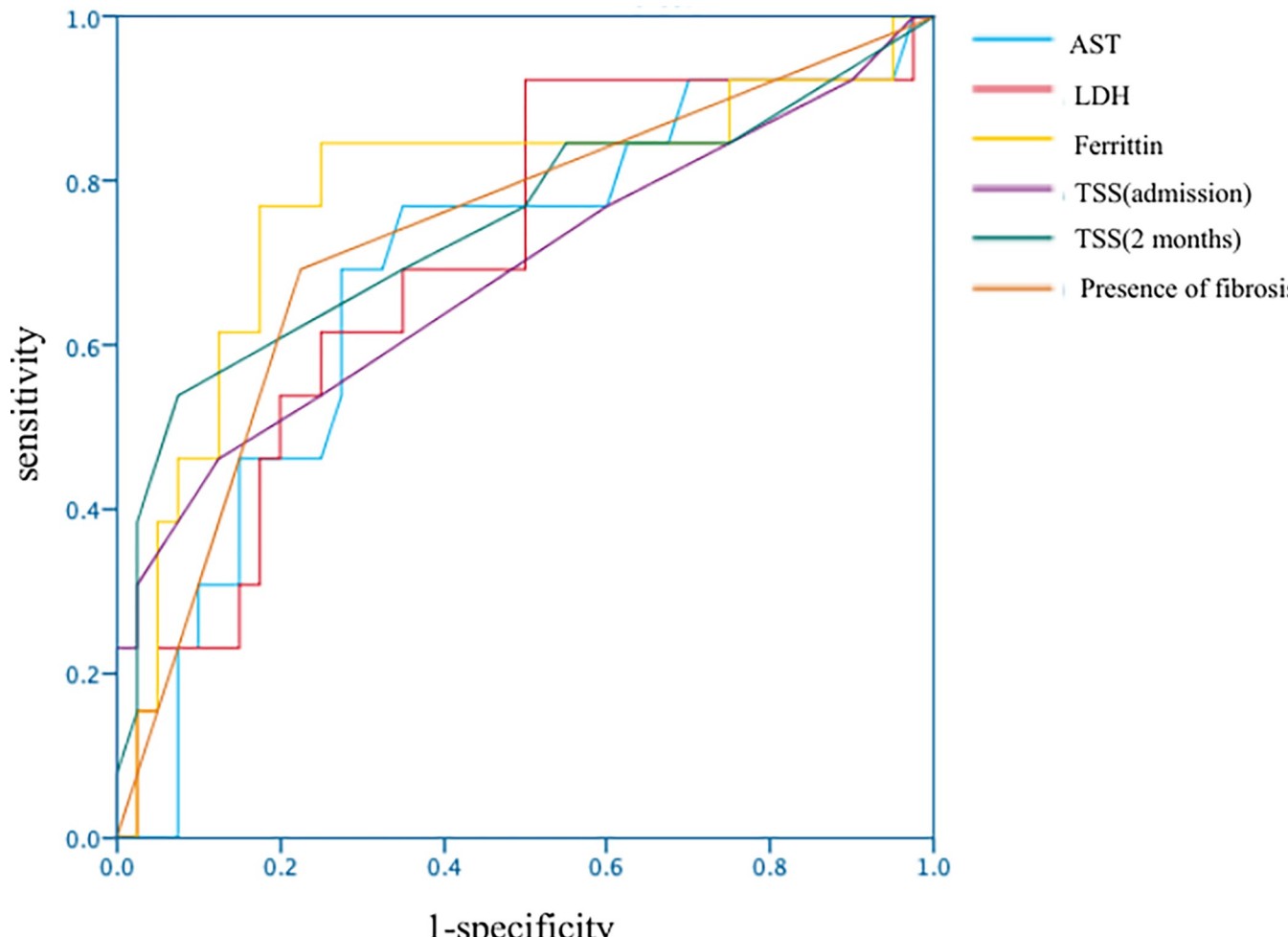

**Fig 1. Receiver operating characteristic curves of clinical factors affecting DLCO impairment in the univariate analysis.** Among the comparison factors, ferritin had the highest area under the curve. DLCO: diffusion capacity for carbon monoxide.

**Table 4. Logistic regression analysis of risk factors for DLCO impairment.**

| Risk factors | Univariate OR (95% CI) | *p*-value | Multivariate OR (95% CI)[a] | *p*-value |
|---|---|---|---|---|
| AST ($> 49.5$ U/L) | 6.43 (1.52-27.21) | 0.011 | ••• | ••• |
| LDH ($> 273.5$ U/L) | 12.6 (1.50-106.02) | 0.020 | ••• | ••• |
| Ferritin ($> 686.5$ ng/mL) | 17.05 (3.22-90.28) | 0.001 | 11.08 (1.84-66.59) | 0.009 |
| TSS at admission ($> 6.5$) | 6.17 (1.47-25.96) | 0.013 | ••• | ••• |

[a] Multivariate analysis adjusted for age, sex, and smoking history.

OR: odds ratio, CI: confidence interval, AST: aspartate aminotransferase, LDH: lactate dehydrogenase, TSS: total severity score.

general malaise. With regard to respiratory dysfunction, DLCO impairment was the most common. The clinical factor most strongly associated with DLCO impairment was the peak ferritin level. In the multivariable analysis, a peak serum ferritin level >686.5 ng/mL was a risk factor for impaired DLCO.

With regard to sequelae symptoms, the high prevalence of dyspnea and general malaise was similar to that in a previous report [4]. The prevalence of residual symptoms was higher in the severe group, which was also similar to that in a previous report [21]. In the present study, only 22% of patients had residual symptoms at 3 months, while 87% of patients in an Italian study had residual symptoms at 2 months and 76% of patients in a Chinese study had residual symptoms at 6 months [4, 21]. Factors contributing to a lower prevalence of sequelae in the present study include the fact that these studies used questionnaires, while the present study was based on interviews during outpatient consultations, and minor or infrequent symptoms may not have been recorded. Then, in our study, all patients were Asian, whereas the Italian study did not provide details but may have included more whites. Racial differences could also contribute to the differences in the proportion of residual symptoms between studies. In addition, the Italian study was from the first wave of the infection, when the efficacy of dexamethasone and antiviral drugs had not been established. Differences in treatment may also have affected residual symptoms.

On pulmonary function tests 3 months after hospital discharge, decreased DLCO was observed in 52% of cases, the most common functional impairment. This finding is consistent with those of previous reports. The percentage of DLCO <80% pred 3 months after hospital discharge has been reported to range from 16% to 82%, depending on disease severity [5, 10, 11]. Furthermore, in this study, the proportion of DLCO <80% pred and DLCO/VA <80% pred was 24%, which showed a stronger correlation with severity than DLCO <80% pred alone, and was thought to be a better indicator of true residual diffusion impairment. In autopsy studies of patients who have died of COVID-19, lung pathology showed diffuse alveolar damage (DAD) findings [22, 23], suggesting that damage to alveolar epithelial cells, basement membrane, and pulmonary capillary endothelium, as well as the associated growth of collagen fibers, occurred in these cases with residual DLCO damage. On the other hand, TLC and FVC decreased with increasing severity, although less frequently (7%) than in DLCO reduction. The finding that TLC decreased in a small number of severe cases is consistent with those of previous studies [9, 10], suggesting that the destruction of alveolar structures by fibrosis associated with DAD and residual atelectasis may be responsible for the decrease.

In a multivariable analysis of clinical factors, peak ferritin level was the factor most strongly associated with DLCO impairment. Apart from COVID-19, there were no cases in this study with a history of autoimmune disease or blood transfusions that contributed to elevated ferritin levels. Previous reports on blood biochemistry have shown an association with the D-dimer level [5]. No other blood biochemical markers associated with respiratory dysfunction

have been reported. Serum ferritin level has already been shown to be associated with COVID-19 severity and mortality [24]. However, to the best of our knowledge, no studies have shown a direct association between elevated ferritin level and respiratory dysfunction. This study is the first to suggest a direct association of ferritin level and DLCO impairment with COVID-19 pneumonia. Ferritin consists of a shell protein with iron molecules encapsulated in the center [25]. Ferritin synthesis is enhanced by the production of cytokines [26]. During cytokine storms, the levels of cytokines, such as interleukin (IL)-1β, IL-6, IL-12, interferon, and tumor necrosis factor-α, are elevated and ferritin is secreted from macrophages, the liver, and Kupffer cells. The uncontrolled virus invades the tissues and infects cells in the alveolar epithelium, airway epithelium, and pulmonary capillary endothelium, causing inflammatory cell death and leading to respiratory dysfunction. To cope with the dead inflammatory cells, an even larger number of immune cells infiltrate into the lungs, presumably resulting in excessive production of inflammatory cytokines [27]. Ferritin is presumed to indirectly reflect this cytokine storm. The association between DLCO impairment and ferritin level suggests that elevated ferritin may indicate inflammatory cell death associated with respiratory dysfunction. In this study, a peak serum ferritin level of >686.5 ng/mL was a strong risk factor for DLCO impairment, and careful follow-up examination and rehabilitation intervention may be effective in preventing residual respiratory disability in patients with this finding [28]. It is also possible that elevated ferritin level could be used as a predictor of respiratory dysfunction due to viral pneumonia.

There are limitations to this study. First, it was a retrospective study with a limited number of cases at a single institution. The applicants were included and selection bias is taken into account. A larger prospective study of clinical factors predicting respiratory dysfunction after COVID-19 is needed. Second, there were no pulmonary function tests conducted before the onset of COVID-19. Cases of respiratory diseases associated with DLCO impairment, such as chronic obstructive pulmonary disease and interstitial pneumonia, were not included. Nevertheless, it could not be ascertained whether there was already reduced DLCO before the onset of COVID-19.

## Conclusions

DLCO impairment was the most frequent disorder of respiratory function, and the peak serum ferritin level was associated with DLCO impairment. A ferritin level of >686.5 ng/mL could be a risk factor for DLCO impairment and indirectly reflects tissue damage with cytokine storms.

## Acknowledgments

We thank Dr. Akiyoshi Kinoshita, Department of Gastroenterology at Jikei University Daisan Hospital, for his help with the statistical analysis. We would also thank all the medical staff, including nurses and laboratory technicians, who are involved in the treatment of COVID-19 patients at our hospital.

## Author Contributions

**Conceptualization:** Kyota Shinfuku, Naoki Takasaka, Yoshitaka Seki, Takeo Ishikawa.

**Data curation:** Kyota Shinfuku, Kentaro Chida, Yudai Suzuki, Shun Shibata, Ayako Kojima, Tsukasa Hasegawa.

**Formal analysis:** Kyota Shinfuku, Taiki Fukuda.

**Investigation:** Kyota Shinfuku, Taiki Fukuda, Shun Shibata, Ayako Kojima, Tsukasa Hasegawa, Masami Yamada, Yumie Yamanaka, Yusuke Hosaka.

**Project administration:** Aya Seki, Yoshitaka Seki, Hiroshi Takeda, Takeo Ishikawa.

**Supervision:** Naoki Takasaka, Kazuyoshi Kuwano.

**Visualization:** Kyota Shinfuku.

**Writing – original draft:** Kyota Shinfuku.

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
