## [Decision Letter · Decision Letter 0]

24 Aug 2022

PONE-D-22-09029Association between serum ferritin levels and decreased diffusion capacity 3 months after the onset of COVID-19 pneumoniaPLOS ONE

Dear Dr. shinfuku,

Thank you for submitting your manuscript to PLOS ONE. After careful consideration, we feel that it has merit but does not fully meet PLOS ONE’s publication criteria as it currently stands. Therefore, we invite you to submit a revised version of the manuscript that addresses the points raised during the review process.

The manuscript has been evaluated by two reviewers, and their comments are available below.

The reviewers have raised a number of major concerns, including the small sample size, the lack of baseline data, and the possibility of other causes of elevated ferritin levels. Could you please carefully revise the manuscript to address all comments raised?

We look forward to receiving your revised manuscript.

Kind regards,

Steve Zimmerman, PhD

Associate Editor, PLOS ONE

https://journals.plos.org/plosone/s/file?id=ba62/PLOSOne_formatting_sample_title_authors_affiliations.pdf".

Reviewers' comments:

Reviewer's Responses to Questions

**Comments to the Author**

1. Is the manuscript technically sound, and do the data support the conclusions?

Reviewer #1: Partly

Reviewer #2: Yes

2. Has the statistical analysis been performed appropriately and rigorously? 

Reviewer #1: Yes

Reviewer #2: Yes

3. Have the authors made all data underlying the findings in their manuscript fully available?

Reviewer #1: Yes

Reviewer #2: Yes

4. Is the manuscript presented in an intelligible fashion and written in standard English?

Reviewer #1: Yes

Reviewer #2: Yes

5. Review Comments to the Author

Reviewer #1: Small sample size.

Given the absence of baseline lung function measurement, it may be that 13/54 patients already had abnormal DLCO/ DLCO-Va prior to COVID-19.

Abstract-background should define what’s known in the field or invoke interest to your question rather than describe your methods.

With a sample size of 13, it is difficult to make a conclusion that ferritin can be a long term lung function abnormalities in COVID-19. At best, you can report an association.

The manuscript presents no novel findings. How does this paper add to the scientific literature?

Please better clarify CT shadows in introduction. What is CT shadows?

Standard definition of severe and critical COVID-19 has not been used.

How did you decide the blood test results selected? Was it just the data that was available?

If you used CT findings to define your outcomes, mention this in the abstract.

Alternative causes of DLCO reduction were not ruled out.

Present findings in table 2. Table 1 should be demographics.

Please correct typing errors.

Reviewer #2: Though this is a small study, it highlights an important association between ferritin and diffusion capacity in COVID19. It serves to identify patients who may be at a higher risk for developing future interstitial lung disease and also aids in those who may benefit from post COVID PFT surveillance.

Some questions from my review as below:

1) Could not understand how AST, LDH and TSS that were significant in univariate analysis, did not show significance in multivariate analysis? Can explain the reason for this as Table 3 and Table 4 seem to show significant p-values for them for AUC and OR respectively. Is this related to the number of patients?

2) The relationship between DLCO and Ferritin is well documented in conditions such as Thalassemia especially when transfusion is needed. Were there any patients who were transfused? Also were there any underlying inflammatory diseases in the patients such as arthritis, autoimmune conditions that could have elevated the ferritin level?

6. PLOS authors have the option to publish the peer review history of their article (what does this mean?). If published, this will include your full peer review and any attached files.

Reviewer #1: No

Reviewer #2: **Yes: **Bright Thilagar

---

## [Author Response · Author response to Decision Letter 0]

15 Oct 2022

Reviewer 1

 The authors would like to thank the reviewer for their constructive critique to improve the manuscript. We have made every effort to address the issues raised and to respond to all comments. The revisions are indicated in blue font in the revised manuscript. Please, find next a detailed, point-by-point response to the reviewer's comments. We hope that our revisions will meet the reviewer’s expectations.

・Small sample size.

We appreciate the reviewer’s comment on this point. We agree with the reviewer that the sample size was small. However, we believe that it is important to note that the frequency of DLCO impairment was quite high. Based on the reviewer’s comments, we have discussed this issue as a limitation as follows:

“There are some limitations in this study. First, it was a retrospective study with a limited number of cases at a single institution.” (Lines 294–295) 

・Given the absence of baseline lung function measurement, it may be that 13/54 patients already had abnormal DLCO/ DLCO-Va prior to COVID-19.

We appreciate the reviewer’s comment on this point. Based on the reviewer’s comments, we have discussed this issue as a limitation as follows:

“Second, there were no pulmonary function tests conducted before the onset of COVID-19. Cases of respiratory diseases associated with DLCO impairment, such as chronic obstructive pulmonary disease and interstitial pneumonia, were not included. Nevertheless, it could not be ascertained whether there was already reduced DLCO before the onset of COVID-19.” (Lines 295–299)

・Abstract-background should define what’s known in the field or invoke interest to your question rather than describe your methods.

We appreciate the reviewer’s comment on this point. Per the reviewer’s insightful suggestion, we have added more background information in the Abstract section as follows:

“The clinical factors associated with DLCO impairment, including blood biochemistry test parameters, remain unclear.” (Lines 23–24)

・With a sample size of 13, it is difficult to make a conclusion that ferritin can be a long term lung function abnormalities in COVID-19. At best, you can report an association.

 We appreciate the reviewer’s comment on this point. Based on the reviewer’s suggestion, we have revised the conclusion in the Abstract and in the main text as follows:

“Decreased DLCO was the most common respiratory function impairment, and ferritin level was a significantly associated clinical factor. Serum ferritin level could be used as a predictor of DLCO impairment in cases of COVID-19 pneumonia.” (Lines 39–41)

“DLCO impairment was the most frequent disorder of respiratory function, and the peak serum ferritin level was associated with DLCO impairment. Ferritin level of >686.5 ng/mL could be a risk factor for DLCO impairment and indirectly reflects tissue damage with cytokine storms.” (Lines 302–305)

・The manuscript presents no novel findings. How does this paper add to the scientific literature?

We appreciate the reviewer’s comment on this point. It is known that DLCO impairment is common in COVID-19 pneumonia and that ferritin is associated with severity and mortality. However, to our knowledge, this is the first study to show a direct association between the peak ferritin level and DLCO impairment. We believe that this finding is novel and would contribute to literature. We have highlighted the novelty of the study as follows:

“To the best of our knowledge, no other blood biochemical markers associated with respiratory dysfunction have been reported to date. Moreover, the serum ferritin level has already been shown to be associated with COVID-19 severity and mortality [24]. However, to the best of our knowledge, no studies to date have shown a direct association between the elevated ferritin level and respiratory dysfunction. This study is the first to suggest a direct association of ferritin level and DLCO impairment with COVID-19 pneumonia.” (Lines 277–283) 

“It is also possible that elevated ferritin levels could be used as a predictor of respiratory dysfunction due to viral pneumonia.” (Lines 291–293)

・Please better clarify CT shadows in introduction. What is CT shadows?

 We appreciate the reviewer’s comment on this point. Based on the reviewer’s comments, we have provided a description in the Introduction section as follows:

 “Severity of COVID-19 and the spread of abnormal chest shadows on computed tomography (CT) scans in COVID-19 pneumonia cases have been reported to be associated with DLCO reduction [12-14].” (Lines 67–69)

・Standard definition of severe and critical COVID-19 has not been used.

 We appreciate the reviewer’s comment on this point. As the reviewer pointed out, we have revised the severity classification according to the NIH criteria. Based on the reviewer’s comments, we have corrected the description of the study design as follows: 

“The severity was defined according to the COVID-19 treatment guidelines of the National Institute of Health [15]. In this study, patients were classified into three groups according to the aforementioned guidelines. Groups 1, 2, and 3 included patients who were hospitalized but did not require oxygen supplementation; those who were hospitalized and required conventional oxygen; and those who were hospitalized and required high-flow nasal cannula, noninvasive ventilation, invasive mechanical ventilation, or extracorporeal membrane oxygenation, respectively.” (Lines 80–87)

Moreover, we have corrected the description in Table 1. 

・How did you decide the blood test results selected? Was it just the data that was available?

 We would like to thank the reviewer for the questions. To respond to the reviewer’s question, we have added the following part to the revised manuscript:

 “Blood tests were performed on admission and periodically thereafter throughout hospitalization. Then, we evaluated the blood sample data during the hospitalization period that may be related to the severity of COVID-19.” (Lines 92–94) 

Moreover, we have provided more information concerning the frequency of blood test (days apart) in Table 1. 

・If you used CT findings to define your outcomes, mention this in the abstract.

 Per the reviewer’s insightful suggestion, we have added the following part to the Abstract:

 “Clinical factors, including blood test parameters and abnormal chest shadows on computed tomography, of COVID-19 pneumonia associated with DLCO impairment were investigated.” (Lines 27–29)

・Alternative causes of DLCO reduction were not ruled out.

According to the reviewer’s insightful suggestion, we have added the following part to the Discussion section:

“Second, there were no pulmonary function tests conducted before the onset of COVID-19. Cases of respiratory diseases associated with DLCO impairment, such as chronic obstructive pulmonary disease and interstitial pneumonia, were not included. Nevertheless, it could not be ascertained whether there was already reduced DLCO before the onset of COVID-19.” (Lines 295–299)

・Present findings in table 2. Table 1 should be demographics.

We appreciate the reviewer’s comment on this point. Please note that we have made corrections in Tables 1 and 2. The revised parts are presented in blue font.

・Please correct typing errors.

We would like to apologize to the reviewer for the typing errors. Please note that we have double-checked and corrected the typing in the text and tables.

Reviewer 2

 The authors would like to thank the reviewer for their constructive critique to improve the manuscript. We have made every effort to address the issues raised and to respond to all comments. The revisions are indicated in red font in the revised manuscript. Please, find next a detailed, point-by-point response to the reviewer's comments. We hope that our revisions will meet the reviewer’s expectations.

1) Could not understand how AST, LDH and TSS that were significant in univariate analysis, did not show significance in multivariate analysis? Can explain the reason for this as Table 3 and Table 4 seem to show significant p-values for them for AUC and OR respectively. Is this related to the number of patients?

We appreciate the reviewer’s comment on this point. As the reviewer indicated, the small sample size may have had an impact. Moreover, in the present study, multivariate analysis was performed for significant items in univariate terms, with cutoff values set using the ROC curves. The multivariate analysis used a stepwise approach, but only ferritin, which is most associated with DLCO impairment, remained as a variable. AST, LDH, and TSS, which were significant in the ROC curve AUC and univariate analyses, were excluded and did not remain. We thought that AST and LDH would not be significant in the multivariate analysis because of confounding factors, such as liver injury and muscle destruction. We also considered that TSS was removed and did not remain as a final factor because of the stronger association of ferritin with DLCO impairment.

2) The relationship between DLCO and Ferritin is well documented in conditions such as Thalassemia especially when transfusion is needed. Were there any patients who were transfused? Also were there any underlying inflammatory diseases in the patients such as arthritis, autoimmune conditions that could have elevated the ferritin level?

We would like to thank the reviewer for the comment. We again checked the transfusion history of the patients and comorbidities that may have contributed to the elevated ferritin level, such as autoimmune diseases, but none of the cases were applicable. Please note that we have discussed this issue in the revised manuscript as follows: “Apart from COVID-19, there were no cases in this study with a history of autoimmune disease or blood transfusions that contributed to elevated ferritin levels.” (Lines 274–276)

---

## [Decision Letter · Decision Letter 1]

22 Nov 2022

PONE-D-22-09029R1Association between serum ferritin levels and decreased diffusion capacity 3 months after the onset of COVID-19 pneumoniaPLOS ONE

Dear Dr. shinfuku,

Thank you for submitting your manuscript to PLOS ONE. After careful consideration, we feel that it has merit but does not fully meet PLOS ONE’s publication criteria as it currently stands. Therefore, we invite you to submit a revised version of the manuscript that addresses the points raised during the review process.

ACADEMIC EDITOR:

Dear Authors

the paper is extremely interesting and has generated an interesting debate among the reviewers. There are differing opinions on the subject; in this context of debate and extremely conflicting opinions between reviewers, I personally consider the paper worthy of note, although I think it needs to be further refined and clarified.

In particular, taking into account the suggestions of reviewer #3, I would ask you to improve the methodological precision described in the M&M session, particularly in the first part where you describe the groups, explaining well the whole methodology of the construct of your paper. 

Again, personally I have two more considerations. First, as you also suggested in the Discussion, it is possible that the differences you found with respect to the Italian literature are also related to racial differences; however, nowhere do I find an indication of the percentage of the Asian, Caucasian, black population that you analysed. This data needs to be implemented. Secondly, it is also possible that the difference between the Italian study and yours is linked to a different pharmacological treatment (steroids, anticoagulants, anti-IL6, etc.) which had not yet been implemented at the time of the first wave. This consideration must also be implemented too.

I therefore ask you again for more clarity in the methodology (especially in the first part of the M&M session) and the resolution of these two points suggested before.

We look forward to receiving your revised manuscript.

Kind regards,

Samuele Ceruti

Academic Editor

PLOS ONE

Journal Requirements:

Reviewers' comments:

Reviewer's Responses to Questions

**Comments to the Author**

1. If the authors have adequately addressed your comments raised in a previous round of review and you feel that this manuscript is now acceptable for publication, you may indicate that here to bypass the “Comments to the Author” section, enter your conflict of interest statement in the “Confidential to Editor” section, and submit your "Accept" recommendation.

Reviewer #2: All comments have been addressed

Reviewer #3: (No Response)

2. Is the manuscript technically sound, and do the data support the conclusions?

Reviewer #2: Yes

Reviewer #3: No

3. Has the statistical analysis been performed appropriately and rigorously? 

Reviewer #2: Yes

Reviewer #3: Yes

4. Have the authors made all data underlying the findings in their manuscript fully available?

Reviewer #2: Yes

Reviewer #3: Yes

5. Is the manuscript presented in an intelligible fashion and written in standard English?

Reviewer #2: Yes

Reviewer #3: Yes

6. Review Comments to the Author

Reviewer #2: Suggestions were accepted and corrections made. Authors were responsive to the corrections in a reasonable and responsive manner.

Reviewer #3: I was asked to review this revised manuscript in its current form, having not reviewed the initial version.

Overall, I agree with prior reviews in general. This retrospective review of a single institution’s experience reveals that peak serum ferritin measures at the time of admission are the most predictive of a reduced DLCO and DLCO/VA at 3 months. I need to reiterate that the number of subjects is small, and the general lack of clinical characterization of the cohort make the results less than convincing. And there is no description of how these subjects were enrolled. The authors state that the CT was done 2 months after onset, and PFTs were done 3 months after onset. Why? And how can you be so sure that the CTs were done exactly 2 months after and the PFTs done exactly 3 months after? There is no mention of a protocol. There is no discussion of why ferritin levels would correlate with a reduction in DLCO. What mechanism of action would that be? The authors are encouraged to develop a prediction model, using multiple clinical, lab, and radiologic characteristics that could be validated prospectively. But as it stands now, I’m not convinced this adds much to the existing literature regarding reduced DLCO after Covid.

7. PLOS authors have the option to publish the peer review history of their article (what does this mean?). If published, this will include your full peer review and any attached files.

Reviewer #2: **Yes: **Bright Thilagar

Reviewer #3: No

---

## [Author Response · Author response to Decision Letter 1]

9 Dec 2022

Academic editor

 The authors would like to thank the editor for their constructive critique to improve the manuscript. We have made every effort to address the issues raised and to respond to all comments. The revisions are indicated in blue font in the revised manuscript. Revisions that are common to the corrections to the reviewer's remarks are noted in green font. Please, find next a detailed, point-by-point response to the editor's comments. We hope that our revisions will meet the reviewer’s expectations.

・In particular, taking into account the suggestions of reviewer #3, I would ask you to improve the methodological precision described in the M&M session, particularly in the first part where you describe the groups, explaining well the whole methodology of the construct of your paper. 

We appreciate the reviewer’s comment on this point. We added a description of the research methods in the study design section as follows: “Enrolled patients were those who were hospitalized with COVID-19 pneumonia during the period and who requested outpatient follow-up after the sequelae symptoms and respiratory dysfunction were explained to them. From the standpoint of infection control, an outpatient examination and CT scan were performed 2 months later and it was confirmed that the patient was improving. Then, pulmonary function tests were performed 3 months later.” (Lines 81-86)

We then added the time to the examination for each group in Tables 1 and 2. 

Again, personally I have two more considerations. First, as you also suggested in the Discussion, it is possible that the differences you found with respect to the Italian literature are also related to racial differences; however, nowhere do I find an indication of the percentage of the Asian, Caucasian, black population that you analysed. This data needs to be implemented.

We appreciate the reviewer’s comment on this point. We have added a description of the race of the patients in Table 1. In addition, based on the editor’s comments, we have corrected the description of the discussion section as follows: “Then, in our study, all patients were Asian, whereas the Italian study did not provide details but may have included more whites. Racial differences could also contribute to the differences in the proportion of residual symptoms between studies.” (Line 264-267)

 Secondly, it is also possible that the difference between the Italian study and yours is linked to a different pharmacological treatment (steroids, anticoagulants, anti-IL6, etc.) which had not yet been implemented at the time of the first wave. This consideration must also be implemented too.

We appreciate the reviewer’s comment on this point. We have added a description in the discussion section as follows: “In addition, the Italian study was from the first wave of the infection, when the efficacy of dexamethasone and antiviral drugs had not been established. Differences in treatment may also have affected residual symptoms.” (Lines 267-269)

Reviewer 3

 The authors would like to thank the reviewer for their constructive critique to improve the manuscript. We have made every effort to address the issues raised and to respond to all comments. The revisions are indicated in red font in the revised manuscript. Revisions that are common to the corrections to the editor's remarks are noted in green font. Please, find next a detailed, point-by-point response to the reviewer's comments. We hope that our revisions will meet the reviewer’s expectations.

Overall, I agree with prior reviews in general. This retrospective review of a single institution’s experience reveals that peak serum ferritin measures at the time of admission are the most predictive of a reduced DLCO and DLCO/VA at 3 months. I need to reiterate that the number of subjects is small, and the general lack of clinical characterization of the cohort make the results less than convincing. 

And there is no description of how these subjects were enrolled. The authors state that the CT was done 2 months after onset, and PFTs were done 3 months after onset. Why? And how can you be so sure that the CTs were done exactly 2 months after and the PFTs done exactly 3 months after? There is no mention of a protocol. 

We appreciate the reviewer’s comment on this point. We added a description of how the eligible patients were enrolled and how the CT scan was performed after 2 months and the pulmonary function test after 3 months in the study design section as follows: “Enrolled patients were those who were hospitalized with COVID-19 pneumonia during the period and who requested outpatient follow-up after the sequelae symptoms and respiratory dysfunction were explained to them. From the standpoint of infection control, an outpatient examination and CT scan were performed 2 months later and it was confirmed that the patient was improving. Then, pulmonary function tests were performed 3 months later.” (Lines 81-86)

We then added the time to the examination for each group in Tables 1 and 2.

There is no discussion of why ferritin levels would correlate with a reduction in DLCO. What mechanism of action would that be?

We appreciate the reviewer’s comment on this point. We added a description of the association between ferritin levels and DLCO reduction in the discussion section as follows: 

“The uncontrolled virus invades the tissues and infects cells in the alveolar epithelium, airway epithelium, and pulmonary capillary endothelium, causing inflammatory cell death and leading to respiratory dysfunction. To cope with the dead inflammatory cells, an even larger number of immune cells infiltrate into the lungs, presumably resulting in excessive production of inflammatory cytokines [27]. Ferritin is presumed to indirectly reflect this cytokine storm. The association between DLCO impairment and ferritin level suggests that elevated ferritin may indicate inflammatory cell death associated with respiratory dysfunction.” (Lines 299-306)

 The authors are encouraged to develop a prediction model, using multiple clinical, lab, and radiologic characteristics that could be validated prospectively. But as it stands now, I’m not convinced this adds much to the existing literature regarding reduced DLCO after Covid.

We appreciate the reviewer’s comment on this point. We added a corresponding description in the discussion section as follows: “First, it was a retrospective study with a limited number of cases at a single institution. The applicants were included and selection bias is taken into account. A larger prospective study of clinical factors predicting respiratory dysfunction after COVID-19 is needed.” (Line 311-314)

---

## [Decision Letter · Decision Letter 2]

19 Jan 2023

Association between serum ferritin level and decreased diffusion capacity 3 months after the onset of COVID-19 pneumonia

PONE-D-22-09029R2

Dear Dr. shinfuku,

We’re pleased to inform you that your manuscript has been judged scientifically suitable for publication and will be formally accepted for publication once it meets all outstanding technical requirements.

Kind regards,

Samuele Ceruti

Academic Editor

PLOS ONE

Additional Editor Comments (optional):

Dear Authors,

I'm so sorry about this delay in my answer, but as you can read below, there was a lot of debate at the reviewers' level; although one of the reviewers requested that the paper be rejected, I feel that the reasons given are not sufficient to reject the paper, especially, after the authors have taken care of the initial comments.

Reviewers' comments:

Reviewer's Responses to Questions

**Comments to the Author**

1. If the authors have adequately addressed your comments raised in a previous round of review and you feel that this manuscript is now acceptable for publication, you may indicate that here to bypass the “Comments to the Author” section, enter your conflict of interest statement in the “Confidential to Editor” section, and submit your "Accept" recommendation.

Reviewer #2: All comments have been addressed

Reviewer #3: (No Response)

2. Is the manuscript technically sound, and do the data support the conclusions?

Reviewer #2: Yes

Reviewer #3: Partly

3. Has the statistical analysis been performed appropriately and rigorously? 

Reviewer #2: Yes

Reviewer #3: Yes

4. Have the authors made all data underlying the findings in their manuscript fully available?

Reviewer #2: Yes

Reviewer #3: Yes

5. Is the manuscript presented in an intelligible fashion and written in standard English?

Reviewer #2: Yes

Reviewer #3: Yes

6. Review Comments to the Author

Reviewer #2: Although not perfect and from a single institution, I think this data would add to the existing literature on supporting follow up DLCO in post COVID patients and is important in Long COVID research.

Reviewer #3: Thank you for revising the manuscript according to my comments. Upon re-review of the manuscript, you have addressed mostly the concerns that I had upon first review. However, your finding that low DLCO is the most common PFT abnormality is not new, and this small study does not add anything to the established literature on the topic.

7. PLOS authors have the option to publish the peer review history of their article (what does this mean?). If published, this will include your full peer review and any attached files.

Reviewer #2: **Yes: **Bright Thilagar

Reviewer #3: No

---

## [Editor Report · Acceptance letter]

5 Feb 2023

PONE-D-22-09029R2 

Association between serum ferritin level and decreased diffusion capacity 3 months after the onset of COVID-19 pneumonia 

Dear Dr. Shinfuku:

I'm pleased to inform you that your manuscript has been deemed suitable for publication in PLOS ONE. Congratulations! Your manuscript is now with our production department. 

Kind regards, 

on behalf of

Dr. Samuele Ceruti 

Academic Editor

PLOS ONE